# The Relationship between Slack Resources, Resource Bricolage, and Entrepreneurial Opportunity Identification—Based on Resource Opportunity Perspective

**Yongbo Sun, Shuang Du and Yixin Ding ***

Business School, Beijing Technology and Business University, Beijing 100048, China; sunyb@th.btbu.edu.cn (Y.S.); 1830601053@st.btbu.edu.cn (S.D.)

**\*** Correspondence: 10011316553@st.btbu.edu.cn

**Abstract:** There are a lot of slack resources in a company. It is vitally important for an enterprise to use slack resources to identify entrepreneurial opportunities to establish company sustainable development. On the basis of the resource orchestration theory and from resource-opportunity perspective, this paper constructs a framework of slack resources and entrepreneurial opportunity identification, exploring the mediating of resource bricolage, the moderating of network ties, and the moderated mediation of network ties. In our analyses, we used data from companies in eastern China, and statistical hypotheses were validated through a structural equation model with data using the statistical software Amos version 20, SPSS version 22. The research results show that: (1) Absorbed slack resources and unabsorbed slack resources have a positive impact on entrepreneurial opportunity identification. (2) Slack resources indirectly affect the opportunity identification through the mediating role of resource bricolage. Among them, resource bricolage has a fully mediating role between absorbed slack resources and entrepreneurial opportunity identification, and it has a partial mediating role between unabsorbed slack resources and entrepreneurial opportunity identification. (3) Business ties positively moderate the relationship between two types of slack resources and entrepreneurial opportunity identification, and business ties moderate the mediation effect of resource bricolage. The resource-opportunity perspective answers how decisions are made, and the entire model process answers how to create sustainable value (entrepreneurship opportunities). The study guides managers on how to integrate and use external and internal resources, coordinate resource elements, and identify profitable business opportunities.

**Keywords:** resource-opportunity; slack resources; resource bricolage; entrepreneurial opportunity identification; resource orchestration theory; network ties

---

## 1. Introduction

As a company strategy maker, entrepreneurs' response to external risks is crucial to the company's sustainable development. The focus of entrepreneurial research is to answer "how, and by whom to make opportunities to generate future goods and services discovered, assessed, and utilized by entrepreneurs" [1]. Identifying the right business opportunity is one of the most important competencies of successful entrepreneurs. Explaining the cognitive process of opportunity is a key part of entrepreneurial research [2]. Research on entrepreneurial opportunities emphasizes the important role of resources on opportunity. The discovery, integration, and utilization of resource elements by enterprises are of great significance for opportunity assessment and opportunity identification of enterprises [3,4]. However, some enterprises have a certain resource, and the allocation of internal

resource elements is unreasonable. They cannot convert the restructured resources into opportunity value, and it is difficult to have a sustainable development. Therefore, it is of great practical significance for enterprises to successfully construct a path from the discovery of resource elements, to the integration and combination of resources, and to the transformation of resources into opportunity value.

Regarding research on the role of bricolage on opportunity, prior studies mainly focus on the resource-based theory and resource dependence theory. The resource-based view emphasizes the heterogeneous resources that are unique and are not easily acquired by other competitors. It is the main source of competitive advantage [5]. However, this view ignores the interaction between enterprises and other entities in their relationship network. Although resource dependence theory believed that companies rely on external resources and have positive interactions with the outside [6], it does not explain the ability of companies to moderate the dependence and independence. To make up for the research gap, this paper introduces the resource allocation theory [7] and combines with the internal resource elements and external interactions to study the impact of bricolage on opportunity identification. The resource orchestration theory not only pays attention to the acquisition process of external resources, but also pays attention to the internal resource construction process, emphasizing the entrepreneurial initiative. From a process perspective, resource orchestration theory [8] includes three stages of enterprise resource structuring, bundling, and leveraging, expounding the path from adaptation of internal and external resources, to integrating and reorganizing resources, to converting resources into opportunity identification.

By sorting out the previous entrepreneurial literature, the research on the cause of entrepreneurial opportunity identification mainly focuses on previous experience, individual characteristics of individuals, entrepreneurial behavior, and social networks [9–12]. Some scholars [13] pointed out that the existence of slack resources enables enterprises to reduce the operating tension of enterprises in market competition, and the impact of the external environment, so enterprises can able to search for information and opportunities more freely in the market. What is more, other scholars found that slack resources could be a source of agency problems, which lead to low efficiencies or inefficiencies of managers, so managers rarely focus on identifying new opportunities. Faced with these controversial research conclusions, it is necessary to explore the impact of slack resources on entrepreneurial opportunity identification through the identification of underlying mechanisms. As a specific resource learning and resource combination method [14,15], resource bricolage can provide new insight about how to convert different resource combinations into new opportunity value.

Further, if entrepreneurs want to perceive and discover potential business opportunities, building network ties for market information is extremely important. Entrepreneurs in the context of social networks often interact with other members for their own goals [16]. In the process of interaction, entrepreneurs obtain financing, information, and motivation resources, which provide important conditions for entrepreneurs to identify entrepreneurial opportunities [17]. Additionally, the development and utilization of slack resources in the context of social networks have changed. By building a network of relationships, entrepreneurs often realize entrepreneurial learning through the sharing of entrepreneurial information and knowledge [18] and finally acquire new methods for developing slack resources. Finally, through online activities and personal interactions, entrepreneurs not only establish social connections with business participants, but also establish social connections with government officials. Therefore, this paper divides the network ties into business ties and political ties, and explores the different impact mechanisms of slack resources to opportunity identification under different network ties.

First, to make up for the shortcomings of the resource-based theory and the resource-dependent theory, this paper introduces the resource orchestration theory to study the value created by the creative integration of internal and external resources. Second, to clarify the relationship between slack resources and entrepreneurial opportunities identification, this paper introduces the intermediary of resource bricolage to reveal the influence mechanism between slack resources and entrepreneurial opportunities identification. Finally, this paper introduces the moderating variable of network ties to explore the changes in the influence mechanism in different situations. Through data analysis,

we found that slack resources have a positive impact on entrepreneurial opportunity identification. Slack resources indirectly affect the opportunity identification through the mediating role of resource bricolage. Business ties positively moderate the relationship between two types of slack resources and entrepreneurial opportunity identification, and business ties moderate the mediation effect of resource bricolage.

This paper considers that the decision to deal with risk is to seek risk, that is, to find new opportunities. In the short term, it may not bring benefits. From a long-term and sustainable perspective, new entrepreneurial opportunities can bring value premiums. It depends on the company's behavioral decisions on entrepreneurial opportunities. From the perspective of resources-opportunity, this paper reveals the process mechanism for enterprises to determine the type of resources, integrate resources, and give value to resources by resource structuring, bundling, and leveraging. This will answer the implementation process and result value of corporate behavioral decisions.

The theoretical contribution of this paper mainly includes two points: First, this paper introduces resource orchestration theory to dynamically explain the link between the existing resources elements—integration and combination of resources—utilization and transformation of resources, which enrich the resource opportunity perspective. Second, previous studies have argued that resource bricolage is a way of resource integration when enterprises face resource constraints. This paper expands this scope and believes that enterprises with slack resources will identify opportunity by the way of bricolage. And network ties strengthen the influential mechanism, which expands the research scenarios and boundaries of resource bricolage, enriching bricolage theory.

This paper is structured as follows. In Section 2, the literature review investigates existing research on resource orchestration theory and entrepreneurial opportunity identification to identify the theoretical gap and hypotheses that were developed. In Section 3, study design and measurement methods are presented. In Section 4, data analysis was performed in four stages, i.e., confirmatory factor analysis, descriptive statistics, common method bias (CMB) testing, and hypotheses testing. In Section 5, we report the general discussion, the theoretical implications, practical implications and limitations, and future research.

## 2. Theory and Hypotheses

### 2.1. Resource Orchestration Theory

Resource-based theory explains the difference in performance due to heterogeneous resources of the enterprise, but the theory only points out the importance of heterogeneous resources and does not explain where the heterogeneous resources of the enterprise come from. To make up for the lack of resource-based view, Sirmon (2007) [8] proposed the resource orchestration theory and constructed a resource orchestration framework, which was defined as the complete process of enterprise resource structuring, bundling, and leveraging. Enterprise resource structuring refers to how companies acquire, accumulate, and integrate resources to find the best combination of internal and external resource elements. Resource bundling refers to the process of how resource elements break the original attributes and recombine to form heterogeneous resources. Resource utilization refers to the process of how the resulting combination of resources is transformed to create new value and form a competitive advantage. Resource orchestration theory has been extended to other areas of management, such as the family business sector [19], and social innovation [20]. The existing research ignores the research on the resource orchestration process. The resource orchestration process involves three aspects: Resource structure, resource bundling, and resource transformation and utilization. At the same time, resource orchestration theory pointed out that the source of competitive advantage lies in not only the resources elements it owns, but also the arrangement of resource elements. The theory emphasizes the effective management and utilization of resource elements. Therefore, this paper constructs a research framework of slack resources–resource bricolage–opportunity identification by introducing resource orchestration theory to link the potential relationship between the acquisition of resource elements,

the integration of resources, and the transformation of resources, which explains the process mechanism of resource orchestration in more detail.

## 2.2. Entrepreneurial Opportunities Identification

We can, with some expansion of earlier expositions, broadly classify three different perspectives: The Discovery View, the Creation View, and the Evolving Idiosyncrasy View. Davidsson (2015) [21] pointed out that these three perspectives have certain limitations and re-conceptualized entrepreneurial opportunities. He believed that entrepreneurial opportunity identification involves external enablers, new venture ideas, and opportunity confidence. Vogel (2017) [22] developed a framework to emphasize that the initial end of entrepreneurial opportunities is identifying opportunities. So, identifying business opportunities is important. Entrepreneurial opportunity is a kind of entrepreneurial activity that occurs from the initial stage to the full maturity stage. Entrepreneurial opportunity is the formation of new means, new goals, or new means–goals to achieve the possibility of introducing new products, new services, new raw materials, and new ways of an organization [1]. At the same time, from a sustainable perspective, some scholars found that identifying, evaluating, and developing sustainable entrepreneurial opportunities can improve the ecological and social environment, and obtain economic returns while eliminating market failure related to ecological and social problems [23–25].

The current literature on entrepreneurial opportunity identification mainly focuses on the role of managers, social networks, and entrepreneurial behaviors [9–12,26]. On the basis of the resource opportunity chain, this paper emphasizes the important role of resources in identifying opportunities. Some scholars have studied the influence mechanism of resource on entrepreneurial opportunity identification [4,27]. However, they only considered resource types and resource bricolage capabilities. Thus, we introduce resource orchestration theory and study the influence mechanism between resource types and resource bricolage capabilities on entrepreneurial opportunity identification, which provides theoretical guidance for managers on how to integrate internal and external resources to promote sustainable corporate development.

## 2.3. Slack Resource and Opportunity Identification

The enterprise growth theory of Penrose and the behavioral theory of Cyert and March both position slack resources as the key to business success, and the slack and excess resources are closely related to the growth and innovation of the enterprise [28]. To theoretically build slack resources as a promoter of entrepreneurial opportunities, this paper draws on the relevant arguments of resource orchestration theory [7], which extends the resource-based theory by transcending resource ownership. The resource orchestration theory must consider the relative availability of slack resources to the enterprise and how resources can be absorbed and transformed into opportunities for entrepreneurs to identify. Specifically, on the basis of resource management and asset orchestration, Sirmon [7] introduced resource orchestration to better understand how companies gain and maintain competitive advantages. Past studies have shown that resources need to be structured, tied, and leveraged, not just with resources, suggesting a competitive advantage when managing resources effectively [29]. This paper argues that the availability of slack resources reflects the advantage of resource orchestration.

Resources "encourage" companies to participate in experiments and explorations related to new product markets and new competitive strategies [30,31], paying attention to perceiving profit opportunities emerging in information asymmetric markets and responding quickly. Because opportunities are essentially price differences, and opportunity profits come from the possibility of arbitrage [31]. The arbitrage process can motivate entrepreneurs to identify and exploit previously unforeseen opportunities [32]. There are slack resources available within the enterprise that may trigger this opportunity identification behavior. This view is supported by empirical evidence from Voss et al. [33]. Absorbed slack resources have increased the likelihood that firms will use existing resources and reduce the exploration of unknown resources. Because absorbed slack resources are often associated with existing product-service-markets, which is equivalent to excess costs in the organization [34], emphasizing

the extent to which the company's engineering operations and product development capabilities are not fully developed. Enterprises explore the company's equipment value, production capacity, and management capabilities to activate the enterprise's operational functions. Additionally, enterprises use unique strategic thinking and skills to re-examine the value and function of the absorbed resources to enhance the identification of opportunities that are matched in undeveloped potential. For example, Uber and Didi take rational advantages of idle vehicles in society, and Airbnb's make creatively use of vacant homes, which has unearthed great commercial value and won the success of entrepreneurship.

By introducing absorbed slack resources into the organization, enterprise can conduct exploration activities to promote the identification of opportunities that are full of uncertainty. In addition, if enterprises have considerable unabsorbed slack resources, they are likely to invest these resources in more promising projects, even if they involve higher risks, because they are highly novel [34]. As enterprises focus on exploring the various projects generated for new knowledge, they can increase the degree of commercialization of existing knowledge that is available and can be applied, and increase the creation of business opportunities [35]. In particular, the introduction of unabsorbed resources, such as surplus cash and bank loans, enables enterprises to shift their horizons to activities that were previously difficult to practice, which can make them explore new models and novel products by changing constraints, and look for opportunities in the process of groping. For example, under the financial support of the Alibaba Group, He Ma Xian Sheng has comprehensively applied technologies such as big data and mobile internet to achieve the optimal matching of people, goods, and fields, and developed a new retail format for retail fields. Thus, we propose the following hypotheses:

**Hypotheses 1a (H1a).** *Absorbed slack resources positively relate to opportunity identification.*

**Hypotheses 1b (H1b).** *Unabsorbed slack resources positively relate to opportunity identification.*

*2.4. The Mediating Role of Resource Bricolage*

Opportunity identification is a cognitive process in which individuals strive to connect seemingly unrelated "points" to obtain new products or services [36]. In the resource opportunity chain, special experiences and knowledge from resource bricolage are important for enterprise to identify entrepreneurial opportunities [37]. As a specific way of learning and resource integration, enterprises that master the ability to bricolage resources have generated a new subjective perception of enterprise resources and have generated new insights and new ideas by reconfiguring different resource elements [15] As a result, resource bricolage helps enterprises expand their accessible "chance collections" to identify new and profitable opportunities.

Resource bricolage can motivate enterprises to form larger "opportunities collections". Through improvisation, trial and error, and resource bricolage, enterprises can provide non-theoretical, experience-based knowledge, especially the connection of seemingly unrelated "points", to gain the idea of developing new products or services [36]. Enterprise can extend the scope of the enterprise's knowledge to identify opportunities because their experiences or knowledge are gained through practice. These opportunities are potential and unobservable, and are not yet recognized by other entrepreneurs. Resource bricolage can encourage enterprises to find more opportunities [38].

Resource bricolage can encourage enterprises to form a more diverse "opportunities collections". Because of the unpredictability of the result of bricolage and the trial and error of the process, resource bricolage often produces subjective, unexpected, implicit knowledge, or heterogeneous services [39]. For example, bricolage may bring subjective perceptions about how resources are reconfigured, which leads to the choice of possible means of meeting new market demands. This new combination of resources helps an enterprise update or recombine resource elements based on unique perceptions of their surroundings. The diversity of ways in which resource elements are recombined can enable the enterprise to identify more opportunities. Moreover, the subjective knowledge generated by the resource bricolage also shapes the unique cognitive framework and model of the enterprise, which

further encourages the enterprise to identify opportunities that can be developed and utilized and have potential [36]. Thus, we propose the following hypothesis:

**Hypotheses 2 (H2).** *Resource bricolage positively relates to opportunity identification.*

Slack resources reduce the barriers by utilizing resources on hand, and resource bricolage creates conditions for opportunity identification. Slack resources not only play a vital role in resolving conflicts and preventing organizational fragmentation, but also create buffers that reduce information processing and coordination costs [40]. When enterprises face resource constraints, the slack resources held by enterprises are like a kind of "cardiotonic agent," which maintains the internal stability of the enterprise, reduces cost and expenditure, and smoothly carries out resource bricolage. Resource bricolage enables entrepreneurs to achieve "from scratch" in the face of resource constraints [41]. In the process of identifying entrepreneurial opportunities, the entrepreneurs often draw funds from weakly related or underdeveloped resources. As a form of value creation, resource bricolage enables entrepreneurs to discover many opportunities that others have not discovered.

On the other hand, slack resources enhance the flexibility and improvisation of the enterprise 's resource bricolage, which helps to better identify entrepreneurial opportunities. Slack resources can ease constraints and improve the strategic choices for entrepreneurial activities, and the availability of resources provides not only protection against threats, but also flexibility to overcome obstacles [33,42]. Since internal slack resources are underutilized resources, allowing enterprises to try and explore, which provides enterprises with enough flexibility and improvisation in the production process, management, and decision-making process, and utilize the skills of bricolage to develop the potential of slack resources. The improvisation of resource bricolage emphasizes the ability of entrepreneurs to generate heterogeneous values from seemingly identical resources [5]. Bricolage tends to utilize resources in a way that is constructively constructed and advocates creativity, improvisation, and novelty seeking, which helps entrepreneurs identify an entrepreneurial opportunity. Thus, we propose the following hypotheses:

**Hypotheses 3a (H3a).** *Resource bricolage mediates the relationship between absorbed resources and opportunity identification.*

**Hypotheses 3b (H3b).** *Resource bricolage mediates the relationship between unabsorbed resources and opportunity identification.*

*2.5. The Moderating Role of Network Ties*

Since economic behavior is rooted in a network of relationships, the perspective of network ties emphasizes the important role of social relations as a coordinated exchange [43]. Through online activities and personal interactions, managers establish not only social connections with business participants, but also social connections with government officials. Business ties are informal social relationships between business and business organizations, such as buyers, suppliers, competitors, and other market partners; political ties are informal social relationships between companies and government officials at all levels, including central and local governments, and agency officials, such as the Tax Administration [44,45]. However, business and political ties are fundamentally different from their role in the resource opportunity path.

Business ties can provide important market resources for enterprise. First, business ties provide important market information that may not be available in open markets, such as subtle changes in related events [46], and partner information [47]. On the basis of obtaining more information about the market, the enterprise expands its exposure to market opportunities with enough slack resources to identify more opportunities from the competitive environment of information asymmetry. Second, close social interaction and communication promote learning and mutual adjustment between business partners, which facilitate knowledge transfer and technology acquisition [48]. By combining new knowledge with its existing knowledge, enterprises can acquire knowledge and skills, and increase the

utilization of redundant resources. They can develop the potential for absorbed slack resources in a continuous interaction process and activate the flexibility of unabsorbed slack resources to discover business opportunities from the creation of resource values. Finally, because past behavior is observable and represents the reputation of the business, network ties can help enterprise gain network legitimacy in the business community [49]. This legitimacy is a strategic resource that attracts business partners and increases the utilization of slack resources. Enterprises maximize the use of limited slack resources under the framework of legality, promote the mutual benefit of all parties, and improve the recognition rate of opportunities in the relationship of trust, commitment, and interdependence.

Political ties help enterprise gain critical regulatory resources. First, the government guides economic activities by formulating industry development plans and formulating regulatory policies. Political ties provide important policy and industry information channels for enterprises. The development of enterprises not only pays attention to the role of the market, but also needs to pay attention to policy guidance and support. Obtaining the latest and most timely policy information can effectively promote the rational use of slack resources by enterprises. Then, enterprises can gain more opportunity in the support of the policy. Second, political ties increase the political legitimacy of enterprises, or government officials, or institutions, and make them believe that the actions of focus enterprises are desirable and appropriate [50]. Enterprises with government ties can gain political legitimacy by gaining positions in government seats [51]. Political legitimacy can help enterprises gain government recognition and support in the process of deploying slack resources. Establishing a relationship with the government not only gains a good reputation, but also facilitates cooperation between the enterprises and relevant research institutions and research institutes. Combining their own resources and the technology of scientific research institutions, enterprises can reduce self-development and bring the risk of failure and cost loss, and encourage long-term relationships, finding profitable opportunities in a political partnership. Thus, we propose the following hypotheses:

**Hypotheses 4a (H4a).** *Business ties moderate the relationship between absorbed slack resources and opportunity identification, such that this relationship is stronger at higher levels of business ties.*

**Hypotheses 4b (H4b).** *Business ties moderate the relationship between unabsorbed slack resources and opportunity identification, such that this relationship is stronger at higher levels of business ties.*

**Hypotheses 4c (H4c).** *Political ties moderate the relationship between absorbed slack resources and opportunity identification, such that this relationship is stronger at higher levels of political ties.*

**Hypotheses 4d (H4d).** *Political ties moderate the relationship between unabsorbed slack resources and opportunity identification, such that this relationship is stronger at higher levels of political ties.*

Business ties can help enterprises establish close collaborative relationships with key business actors and engage in entrepreneurial activities by acquiring the broad knowledge, resources, and complementary capabilities of their partners. Extensive contact with suppliers can provide a wealth of knowledge, gain more ways to solve problems, and find new ways to combine between different elements [52]. Enterprises can improve the ability of resource bricolage and take advantage of the value of slack resources to develop business ideas into a feasible opportunity by enriching the use of creative processes. Similarly, customer contact helps enterprises easily identify new market needs and quickly adapt to market changes to better understand customer preferences and identify new market positions. To meet new market demands and identifying new opportunities, enterprises should effectively expand slack resources and rationally utilize the ability of bricolage. Business ties can help enterprises bring together resources from different partners to achieve scale economies in their entrepreneurial activities [53]. In the process of cooperation, enterprises can acquire the knowledge and resources needed for the main business, and improve the ability of enterprises to better adapt to market changes, solve resource processing problems, and strengthen resource capabilities, to ensure that enterprises can quickly and flexibly identify opportunities in the market.

Since the institutional environment in China is underdeveloped, political ties can compensate for the lack of perfect systems by providing enterprises with access to policy information and valuable resources [45]. Political ties help to strengthen the normative nature of economic exchanges in the market. The government plays a key role in regulating market and organizational behavior, and providing a stable institutional basis through legal or social sanctions [19]. For example, the government develops industry-specific regulations on prices, outputs, and licenses to regulate and guide the proper functioning of market transactions. The legal environment provides enterprises with the ability to integrate and utilize slack resources. It plays and maintains the value of resources that are not suppressed by monopolistic enterprises, and reorganizes resources in a way that generates profits and creates opportunities. Political ties can improve the legitimacy of the enterprises so that suppliers, customers, government agencies, etc., agree with the value of the existence of the enterprise. They are more likely to trust enterprises with high legitimacy and purchase their innovative products or technologies [54]. Specifically, enterprises use slack resources for product development or service creation. This product or service must be approved by society. Political ties provide enterprises with access to government resources that are invaluable for innovative production. These resources play a vital role in innovation and production. They can create uncertainty about results, increase their legitimacy, and thereby enhance the ability to combine with resources, thus identifying profitable opportunities about products or services in an unsaturated market [55]. Thus, we propose the following hypotheses:

**Hypotheses 5a (H5a).** *Business ties moderate the indirect relationship between absorbed slack resources and opportunity by bricolage, such that this indirect relationship is stronger at higher levels of business ties.*

**Hypotheses 5b (H5b).** *Business ties moderate the indirect relationship between unabsorbed slack resources and opportunity by bricolage, such that this indirect relationship is stronger at higher levels of business ties.*

**Hypotheses 5c (H5c).** *Political ties moderate the indirect relationship between absorbed slack resources and opportunity by bricolage, such that this indirect relationship is stronger at higher levels of political ties.*

**Hypotheses 5d (H5d).** *Political ties moderate the indirect relationship between unabsorbed slack resources and opportunity by bricolage, such that this indirect relationship is stronger at higher levels of political ties.*

To sum up, this study explored the mechanisms influencing the impact of slack resource on entrepreneurial opportunity identification and the moderating effects of network ties. Figure 1 presents the theoretical model of this study.

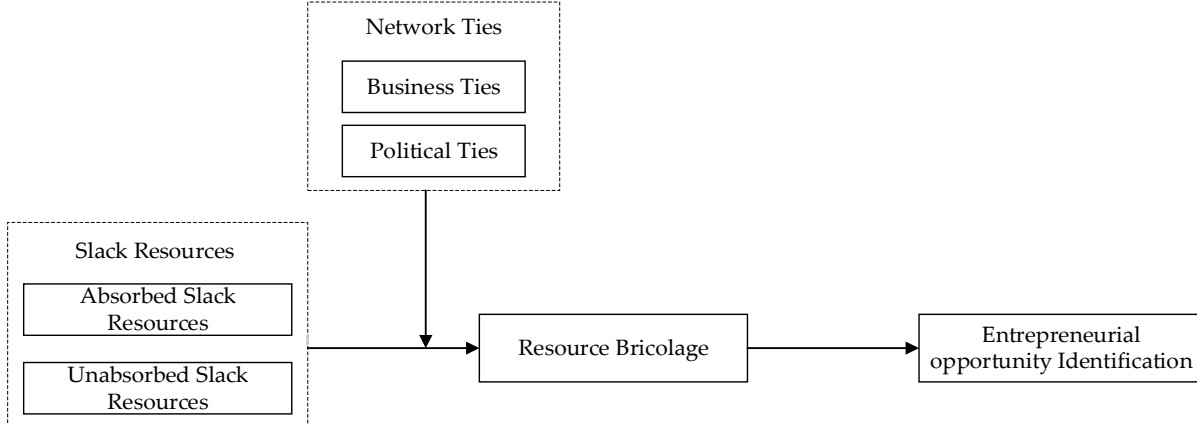

**Figure 1.** Theoretical model. Source: Microsoft Visio drawing.

## 3. Method

### 3.1. Participants and Procedures

The survey areas were mainly concentrated in eastern China, such as Beijing, Shanghai, Hebei province, Tianjin province, Shandong province, Zhejiang province, Guangdong province, and Jiangsu province. Eastern China has a high degree of entrepreneurial activity and various types of enterprises, which meets research requirements.

The survey object of this paper is the top management of the company. To ensure the validity and reliability of the questionnaire content, the two associate professors in the entrepreneurial field and the two senior managers of the enterprise were invited to modify some difficult or ambiguous items before the formal issuance of the questionnaire. After the revision, formal questionnaires were formed. They are issued in two ways: First, we contacted the research company to collect questionnaires from our research subjects. Second, through the mentor project, we contacted the company executives of the project entrusted party, and contacted more through company executives by the way of snowballing.

Because of research costs and sample availability, the study was conducted from March 2018 to July 2018, and lasted for 4 months. We issued a questionnaire through research companies and projects from eastern China. The research company distributed 300 questionnaires and retrieved 201 questionnaires. The recovery rate was 67%; 27 invalid questionnaires were excluded, and 174 valid questionnaires were selected. The effective rate of the questionnaire was 58%. A total of 46 questionnaires were distributed in the project contacting and snowballing methods, 41 were recovered, and the recovery rate was 89.13%; 35 valid questionnaires were selected, and the effective rate of the questionnaire was 76.09%.

This paper descriptively analyzes the sample characteristics of 209 questionnaires. The statistical analysis results are shown in Table 1. As can be seen from Table 1, there is a certain gap between the male and female ratios of the respondents, with men accounting for 59.8%, and women accounting for 40.2%; number of respondents aged between 36 and 45 accounting for 59.3%, is highest. The undergraduate subjects of the respondents were mainly undergraduate, accounting for 78.5%. The industry types were mainly information transportation, software and information technology services, and manufacturing, accounting for 45.9% and 25.8%, respectively. The scale of enterprises was mainly distributed between 101 and 200, accounting for 47.4%. The establishment time of enterprises is mainly distributed between 1 and 3 years, accounting for 45.5%.

**Table 1.** Descriptive statistics.

| Characteristic | Category | Frequency | Percentage | Cumulative Percentage |
|---|---|---|---|---|
| **Sex** | Man | 125 | 59.8 | 59.8 |
| | Female | 84 | 40.2 | 100 |
| **Age** | 25 and below | 5 | 2.4 | 2.4 |
| | 26 to 35 | 73 | 34.9 | 37.3 |
| | 36 to 45 | 124 | 59.3 | 96.6 |
| | 46 and above | 7 | 3.4 | 100 |
| **Education level** | College and below | 5 | 2.4 | 2.4 |
| | Bachelor | 164 | 78.5 | 80.9 |
| | Postgraduate | 40 | 19.11 | 100 |
| **Industry type** | Primary industry | 1 | 0.5 | 0.5 |
| | Manufacturing | 54 | 25.8 | 26.3 |
| | Wholesale and retail trade | 17 | 8.1 | 34.4 |
| | Accommodation | 20 | 9.6 | 44.0 |
| | Information technology | 96 | 45.9 | 90.0 |
| | Transportation and Postal | 19 | 9.1 | 9.0 |
| | Other | 2 | 1.0 | 100 |

**Table 1.** *Cont.*

| Characteristic | Category | Frequency | Percentage | Cumulative Percentage |
|---|---|---|---|---|
| **Scale** | 100 and below | 45 | 21.5 | 21.5 |
| | 101 to 200 | 99 | 47.4 | 68.9 |
| | 201 to 500 | 49 | 23.4 | 92.3 |
| | 500 and above | 16 | 7.7 | 100 |
| **Established time** | 1 year and below | 40 | 19.1 | 19.1 |
| | 1 to 3 | 95 | 45.5 | 64.6 |
| | 4 to 8 | 62 | 29.7 | 94.3 |
| | 8 year and above | 12 | 5.7 | 100 |

### 3.2. Measures

The questionnaire collected ratings using seven-point Likert-type scales ranging from 1 ("strongly disagree") to 7 ("strongly agree"). Items related to variables were translated into Chinese using the standard translation–back translation procedure [56]. Table 2 shows the internal consistency reliability scores for each measure, which are described below.

**Table 2.** Results of confirmatory factor analysis.

| Models | $\chi^2$ | df | $\chi^2$/df | RMSEA | CFI | IFI |
|---|---|---|---|---|---|---|
| **Six factors** | 396.393 | 277 | 1.431 | 0.046 | 0.941 | 0.942 |
| **Five factors** | 466.241 | 284 | 1.642 | 0.056 | 0.910 | 0.912 |
| **Four factors** | 470.095 | 289 | 1.627 | 0.055 | 0.911 | 0.912 |
| **Three factors(a)** | 472.980 | 293 | 1.614 | 0.054 | 0.911 | 0.913 |
| **Three factors(b)** | 817.426 | 293 | 2.790 | 0.093 | 0.741 | 0.745 |
| **Two factors** | 819.022 | 296 | 2.767 | 0.092 | 0.742 | 0.746 |
| **One factors** | 1149.268 | 298 | 3.857 | 0.117 | 0.580 | 0.586 |

Note. (1) AS: Absorbed slack resources, US: Unabsorbed slack resources, BR: Bricolage, OI: Opportunity identification, BT: Business ties, PT: Political ties. (2) One factors: AS + US + BR + OI + BT + PT; Two factors: AS + US + BR + BT + PT, OI; Three factors (b): AS + US + BR + BT, PT, OI; Three factors (a): AS + US + BR + PT, BT, OI; Four factors: AS + US + BR, BT, PT, OI; Five factors: AS + US, BR, PT, BT, OI; Six factors: AS, US, BR, BT, PT.

Slack resources. We drew on the research design of Tan and Peng (2003) [34] and Voss et al. (2008) [33], and modified the scale in combination with the situation in this paper, and finally determined two dimensions and six specific questions.

Bricolage. We used Senyard's [57] eight item scale and modified the scale in combination with the situation in this paper.

Network relationship. We drew on Peng and Luo's (2000) [45] research design combined with the situation in this paper to modify the measurement items, and finally determined 2 dimensions and 6 specific questions.

Opportunity Identification. We drew on the research design of Ozgen and Baron (2007) [58] and Gregoire et al. (2010) [59]. Combined with the scenario of this paper, we modified the measurement items to determined three specific questions.

Control variable. Learning from the research of previous scholars, we selected sex, age, educated level, industry type, enterprise scale, and established time as control variables. Previous studies have shown that the enterprise scale, established time, and industry type will affect the stock of slack resources and network relationships; the managers' sex, age, and educated level will affect the method of bricolage.

## 4. Results

### 4.1. Confirmatory Factor Analysis

We used AMOS 20 to conduct a confirmatory factor analysis of slack resources, bricolage, opportunity identification, and network ties to test the discriminant validity of variables. As shown in

Table 2, the model fit of seven factors ($\chi^2$ = 396.396, df = 277, $\chi^2$/df = 1.431, CFI = 0.941, GFI = 0.881, IFI = 0.942, RMSEA = 0.046) was significantly better than other nested models.

## 4.2. Common Method Bias Testing

In this paper, two statistical methods are used to diagnose CMB. First, the Harman's single-factor test was used to evaluate the CMB, and all variables were put together for exploratory factor analysis. The results showed that the first factor without rotation explained the variance, only accounting for 1.929% of the total variance. No single variable can explain most of the variables, suggesting that CMB may not have a serious impact on the validity of this paper. Second, this paper uses the controlling for the effects of an unmeasured latent methods factor to test the CMB. Based on the six-factor model, the method bias is put into the structural equation as a potential factor to construct a seven-factor model. If the fitting index of the factor model containing the method bias is better than the factor model without the method bias, then the CMB is tested. The results show that the fitting results of the seven-factor model: $\chi^2$/df = 1.374, RMSEA = 0.044, CFI = 0.943, GFI = 0.884, IFI = 0.943.The fitting index of the seven-factor model containing the method bias is better than the factor model without the method bias, and the difference is within 0.003, indicating that there is no serious CMB between the variables in this paper.

## 4.3. Descriptive Statistics

Table 3 presents the descriptive statistics and correlations between all variables. As shown in Table 3, absorbed resources and unabsorbed resources positively correlated with entrepreneurial opportunity identification (r = 0.454, $p$ < 0.010; r = 0.623, $p$ < 0.010). Absorbed resources and unabsorbed resources positively correlated with bricolage (r = 0.604, $p$ < 0.010; r = 0.686, $p$ < 0.010). Bricolage positively correlated with entrepreneurial opportunity identification (r = 0.621, $p$ < 0.010). Business ties positively correlated with entrepreneurial opportunity identification (r = 0.516, $p$ < 0.010).

**Table 3.** Descriptive statistics of the variables.

| Variable | 1 | 2 | 3 | 4 | 5 | 6 | 7 | 8 | 9 | 10 | 11 | 12 |
|---|---|---|---|---|---|---|---|---|---|---|---|---|
| Sex | — | | | | | | | | | | | |
| Age | 0.147 * | — | | | | | | | | | | |
| Education | −0.044 | 0.005 | — | | | | | | | | | |
| Type | −0.110 | −0.099 | 0.054 | — | | | | | | | | |
| Scale | −0.006 | −0.115 | 0.064 | 0.062 | — | | | | | | | |
| Time | 0.086 | 0.053 | −0.001 | −0.084 | −0.021 | — | | | | | | |
| AR | −0.026 | 0.033 | −0.075 | −0.019 | −0.114 | −0.176 * | 0.776 | | | | | |
| UR | −0.055 | −0.042 | 0.027 | −0.111 | −0.110 | −0.050 | 0.424 ** | 0.767 | | | | |
| BR | −0.018 | 0.009 | −0.031 | −0.080 | −0.112 | −0.124 | 0.604 ** | 0.686 ** | 0.788 | | | |
| BT | 0.023 | 0.031 | −0.049 | −0.064 | −0.070 | −0.052 | 0.417 ** | 0.578 ** | 0.654 ** | 0.782 | | |
| PT | −0.101 | 0.008 | 0.094 | 0.081 | 0.094 | 0.011 | −0.048 | 0.027 | 0.016 | 0.043 | 0.772 | |
| OI | 0.015 | 0.004 | −0.076 | −0.070 | 0.103 | −0.112 | 0.454 ** | 0.623 ** | 0.621 ** | 0.516 ** | −0.078 | 0.773 |
| Mean | 5.233 | 5.502 | 5.481 | 5.514 | 0.598 | 2.392 | 5.502 | 5.502 | 5.481 | 5.276 | 5.317 | 5.514 |
| SD | 1.304 | 1.096 | 1.169 | 1.152 | 0.490 | 0.594 | 1.304 | 1.096 | 1.169 | 1.336 | 1.310 | 1.152 |

Notes. ** $p$ < 0.01, * $p$ < 0.05.

## 4.4. Hypotheses Testing

This paper used the multiple regression method of Baron−Kenny (1986) to test the main effect, the mediating effect, and the complementary effect. The hierarchical regression results in Table 4 shows that (1) compared with M3, M4 had a positive impact on entrepreneurial opportunity identification after the influence of fixed control variables (β = 0.212, $p$ < 0.001); β = 0.534, $p$ < 0.001) and can additionally explain the entrepreneurial opportunity recognition variation of up to 40.8% ($\Delta R^2$ = 0.408).

The results show that absorbed slack resources and unabsorbed slack resources had a significant positive impact on entrepreneurial opportunity identification. Hence, H1a and H1b were supported. (2) Compared with model 3, after the influence of model 7 on fixed control variables, the regression coefficient of resource bricolage was significantly positive ($\beta = 0.610$, $p < 0.001$), and an additional 35.9% of ($\Delta R^2 = 0.359$) entrepreneurship opportunity identification can be explained. The results show that resource bricolage had a significant positive impact on entrepreneurial opportunity identification. H2 was supported. (3) Compared with model 4, after the influence of fixed control variables, absorbed slack resources and unabsorbed slack resources, the regression coefficient of resource bricolage was significantly positive ($\beta = 0.288$, $p < 0.001$) and can extra explain 3.4% ($\Delta R^2 = 0.034$) of entrepreneurial opportunity identification. Model 4, model 6, and model 7 provided preliminary support for the mediation test of the resource bricolage.

**Table 4.** Result of main effect.

| Variable | Bricolage | | Entrepreneurial Opportunity Identification | | | | |
|---|---|---|---|---|---|---|---|
| | **M1** | **M2** | **M3** | **M4** | **M5** | **M6** | **M7** |
| **Control** | | | | | | | |
| **Sex** | −0.017 | 0.020 | 0.015 | 0.049 | 0.50 | 0.043 | 0.026 |
| | (−0.246) | (0.424) | (0.218) | (0.910) | (0.921) | (0.829) | (0.462) |
| **Age** | −0.002 | 0.016 | −0.010 | 0.014 | 0.008 | 0.009 | −0.009 |
| | (−0.026) | (0.355) | (−0.138) | (0.258) | (0.142) | (0.175) | (−0.153) |
| **Education** | −0.020 | −0.015 | −0.065 | −0.071 | −0.074 | −0.067 | −0.053 |
| | (−0.290) | (−0.334) | (−0.937) | (−1.338) | (−1.389) | (−1.292) | (−0.957) |
| **Type** | −0.086 | −0.013 | −0.070 | 0.000 | 0.001 | −0.044 | −0.018 |
| | (−1.223) | (−0.281) | (−0.997) | (0.001) | (0.015) | (0.072) | (−0.316) |
| **Scale** | −0.108 | −0.009 | −0.098 | −0.014 | −0.014 | −0.012 | −0.032 |
| | (−1.554) | (−0.189) | (−1.398) | (−0.268) | (−0.259) | (−0.228) | (−0.564) |
| **Time** | −0.132 | −0.040 | −0.121 | −0.055 | −0.057 | −0.044 | −0.040 |
| | (−1.903) | (−0.870) | (−1.730) | (−1.020) | (−1.047) | (−0.828) | (−0.711) |
| **Independent variable** | | | | | | | |
| **AS** | | 0.372 *** | | 0.212 *** | 0.224 *** | 0.105 | |
| **(AS)** | | (7.291) | | (3.559) | (3.674) | (1.616) | |
| **US** | | 0.526 *** | | 0.534 *** | 0.539 *** | 0.382 *** | |
| **(US)** | | (10.391) | | (9.004) | (9.050) | (5.384) | |
| **Mediator variable** | | | | | | | |
| **RB** | | | | | | 0.288 *** | 0.610 *** |
| **(RB)** | | | | | | (3.578) | (10.900) |
| **R²** | 0.036 | 0.593 | 0.034 | 0.442 | 0.455 | 0.476 | 0.393 |
| **ΔR²** | — | 0.557 *** | — | 0.408 *** | 0.013 * | 0.034 *** | 0.359 *** |
| **F** | 1.272 | 36.399 *** | 1.168 | 19.841 *** | 17.723 *** | 20.100 *** | 18.557 *** |

Notes. (1) *** $p < 0.001$, * $p < 0.05$ (2) T value in parentheses.

To further test the median effect, this paper used the mediation effect test procedure proposed by Preacher and Hayes [60], and used the Bootstrap method to test the mediation effect of resource bricolage through the process plugin of SPSS. Mackinnon et al. [61] found that the asymmetric confidence interval method abandons the premise that the sampling distribution of the mediating effect was a normal distribution and did not limit the sampling distribution of the mediating effect. The percentile method of bias correction provided the most accurate confidence interval estimation and reduced the probability of statistical error, and had higher statistical power. In this paper, the Bootstrap sample size was 2000, the confidence interval was set to 95%, and the mediation effect and confidence interval are shown in Table 5.

**Table 5.** The mediation analysis results of Bootstrap.

| Path | | Direct Effect | Indirect Effect | Total Effect |
|---|---|---|---|---|
| **AS→BR→OI** | Effect value | 0.116 | 0.323 *** | 0.439 *** |
| | Confidence interval | [−0.021, 0.253] | [0.222, 0.456] | [0.312, 0.565] |
| **US→BR→OI** | Effect value | 0.386 *** | 0.237 *** | 0.623 *** |
| | Confidence interval | [0.244, 0.527] | [0.117, 0.369] | [0.513, 0.731] |

Notes. (1) *** $p < 0.001$. (2) T value in parentheses. (3) AS: Absorbed slack resources, US: Unabsorbed slack resources, BR: Bricolage, OI: Opportunity identification.

The results of Bootstrap mediation analysis show that at the 95% confidence level, (1) the median direct effect of the resource bricolage between the absorbed slack resources and the entrepreneurial opportunity identification was 0.116, and the confidence interval (LLCI = −0.021, ULCI = 0.253) includes 0, indicating that the resource bricolage variable was added, the effect of absorbed slack resources on the identification of entrepreneurial opportunity was not significant; the indirect effect was 0.323, and the confidence interval (LLCI = 0.222, ULCI = 0.456) did not include 0, indicating that the resource bricolage played a mediating role between absorbed slack resources and entrepreneurial opportunity identification. H3a was supported. It can be seen from the direct effect that resource bricolage played a fully mediating role between absorbed slack resources and entrepreneurial opportunity identification. (2) The direct effect of resource bricolage between unabsorbed slack resources and entrepreneurial opportunity identification was 0.386, and the confidence interval (LLCI = 0.244, ULCI = 0.527) did not include 0, indicating that after the resource bricolage variable was added, the impact of resource bricolage on entrepreneurial opportunity identification was still significant. Its indirect effect value was 0.237, and the confidence interval (LLCI = 0.117, ULCI = 0.369) did not include 0, indicating that resource bricolage played a mediating role between unabsorbed slack resources and entrepreneurial opportunity identification. H3b was supported. It can be seen from the direct effect that resource played partial mediating role between unabsorbed resources and opportunity identification.

In this paper, the multivariate regression method of Baron–Kenny [62] was used to examine the moderating effects (Table 6).

(1) The moderating effect of absorbed slack resources on entrepreneurial opportunity identification. After adding the business ties variable in model 8, the regression coefficient of the business ties was significant (β = 0.341, $p < 0.001$). The results show that the business ties had a significant positive impact on entrepreneurial opportunities identification. Based on model 8, after adding the interaction term of absorbed slack resources and business ties in model 9, the regression coefficient of the interaction term between absorbed slack resources and business ties was positive (β = 0.136, $p < 0.050$). This shows that business ties played a positive moderating role between absorbed slack resources and entrepreneurial opportunity identification. H4a was supported. To test H4a more comprehensively, this paper calculated the simple slope of low business ties and high business relationship by using the mean value of the moderating effects variable (business ties) plus or minus one standard deviation as the grouping criterion. The results show that under low business ties, the simple slope value was −0.071, the t value was −0.328, and the p value was 0.746 > 0.050. Under high business ties, the simple slope was estimated to be 0.178, the t value was 0.811, and the *p* value was 0.426 > 0.050. Thus, the moderating effect diagram of the business ties was drawn. As can be seen from Figure 2: Compared with the low level of business ties, the positive impact of absorbed slack resources on entrepreneurial opportunity identification was stronger; the possibility of high business ties influencing opportunity identification raised faster than the low business ties. In other words, the slope was larger. After adding political tie variables in model 10, the regression coefficient of political ties was not significant (β = −0.045, $p > 0.050$). The results show that political ties had no significant impact on entrepreneurial opportunities identification. Based on model 10, after adding the interaction term of absorbed slack resources and the political ties in model 11, the regression coefficient of the interaction term between

absorbed slack resources and political ties was not significant ($\beta = 0.092$, $p > 0.050$). It shows that the moderating role of business ties between absorbed slack resources and entrepreneurial opportunities identification was not significant. H4b was not supported.

**Table 6.** The result of moderating effect.

| Variable | Entrepreneurial Opportunity Identification | | | | | | | |
|---|---|---|---|---|---|---|---|---|
| | **M8** | **M9** | **M10** | **M11** | **M12** | **M13** | **M14** | **M15** |
| **Control variable** | | | | | | | | |
| **Sex** | 0.012 | 0.012 | 0.020 | 0.019 | 0.037 | 0.037 | 0.041 | 0.039 |
| | (0.197) | (0.208) | (0.316) | (0.296) | (0.674) | (0.696) | (0.730) | (0.698) |
| **Age** | 0.000 | 0.005 | −0.021 | −0.015 | 0.028 | 0.032 | 0.030 | 0.030 |
| | (−0.002) | (0.089) | (−0.323) | (−0.241) | (0.526) | (0.601) | (0.546) | (0.539) |
| **Education** | −0.031 | −0.033 | −0.032 | −0.017 | −0.075 | −0.072 | −0.083 | −0.080 |
| | (−0.532) | (−0.580) | (−0.502) | (−0.270) | (−1.411) | (−1.374) | (−1.512) | (−1.444) |
| **Type** | −0.039 | −0.025 | −0.060 | −0.049 | 0.005 | 0.009 | 0.007 | 0.008 |
| | (−0.665) | (−0.429) | (−0.949) | (−0.762) | (0.087) | (0.167) | (0.124) | (0.138) |
| **Scale** | −0.041 | −0.058 | −0.046 | −0.040 | −0.027 | −0.023 | −0.019 | −0.019 |
| | (−0.700) | (−0.990) | (−0.724) | (−0.635) | (−0.493) | (−0.444) | (−0.350) | (−0.336) |
| **Time** | −0.051 | −0.047 | −0.045 | −0.047 | −0.081 | −0.088 | −0.085 | −0.080 |
| | (−0.864) | (−0.815) | (−0.704) | (−0.735) | (−1.508) | (−1.668) | (−1.551) | (−1.438) |
| **Independent variable** | | | | | | | | |
| **AS** | 0.275 *** | 0.283 *** | 0.437 *** | 0.441 *** | | | | |
| **(AS)** | (4.263) | (4.427) | (6.814) | (6.893) | | | | |
| **US** | | | | | 0.494 *** | 0.518 *** | 0.625 *** | 0.627 *** |
| **(US)** | | | | | (7.494) | (7.936) | (11.332) | (11.313) |
| **Moderating variable** | | | | | | | | |
| **BT** | 0.391 *** | 0.437 *** | | | 0.221 ** | 0.288 *** | | |
| **(BT)** | (6.169) | (6.600) | | | (3.389) | (4.226) | | |
| **PT** | | | −0.045 | −0.048 | | | −0.081 | −0.081 |
| **(PT)** | | | (−0.713) | (−0.759) | | | (−1.476) | (−1.461) |
| **Interaction** | | | | | | | | |
| **AS*BT** | | 0.136 * | | | | | | |
| | | (2.190) | | | | | | |
| **AS*PT** | | | | 0.092 | | | | |
| | | | | (1.440) | | | | |
| **US*BT** | | | | | | 0.171 ** | | |
| | | | | | | (2.879) | | |
| **US*PT** | | | | | | | | 0.027 |
| | | | | | | | | (0.484) |
| **R$^2$** | 0.342 | 0.357 | 0.218 | 0.227 | 0.439 | 0.462 | 0.414 | 0.414 |
| **ΔR$^2$** | — | 0.015 * | — | 0.009 | — | 0.023 ** | — | 0.000 |
| **F** | 12.980 *** | 12.289 *** | 6.989 *** | 6.476 *** | 19.593 *** | 18.972 *** | 17.630 *** | 15.637 *** |

Notes. (1) *** $p < 0.001$, ** $p < 0.01$, * $p < 0.05$. (2) T value in parentheses. (3) AS: Absorbed slack resources, US: Unabsorbed slack resources, BR: Bricolage, OI: Opportunity identification.

(2) The moderating effect of unabsorbed slack resources on entrepreneurial opportunity identification. After adding the business ties variable in model 12, the regression coefficient of the business ties was significant ($\beta = 0.221$, $p < 0.010$). The results show that the business ties had a significant positive impact on opportunity identification. Based on model 12, after adding the interaction term of unabsorbed slack resources and business ties in model 13, the regression coefficient of the interaction term between unabsorbed slack resources and business ties was positive ($\beta = 0.171$, $p < 0.010$). This shows that business ties played a positive moderating role between unabsorbed

slack resources and entrepreneurial opportunity identification. H4c was supported. To test H6c more comprehensively, this paper calculates the simple slope of low business relationship and high business relationship by using the mean value of the regulatory variable business relationship plus or minus one standard deviation as the grouping standard. The results show that under low business ties, the simple slope value is 0.317, the t value is 1.470, and the p value is 0.157 > 0.050. Under high business ties, the simple slope is estimated to be 0.624, the t value is 3.487, and the p value is 0.002 < 0.010. Thus, the moderating effect diagram of the business ties was drawn. As can be shown in Figure 3: Compared with the low level of business ties, the positive impact of unabsorbed slack resources on entrepreneurial opportunity identification was stronger. The possibility of high business ties influencing opportunity identification raised faster than the low business ties. In other words, the slope was larger. After adding political tie variables in model 14, the regression coefficient of the interaction term between unabsorbed slack resources and political ties was not significant. ($\beta = -0.081$, $p > 0.050$). Based on model 14, after adding the interaction term of unabsorbed slack resources and political ties in model 15, the regression coefficient of the interaction term of unabsorbed slack resources and political ties was not significant ($\beta = 0.027$, $p > 0.050$). It shows that the moderating role of political ties between unabsorbed slack resources and opportunity identification. H4d was not supported.

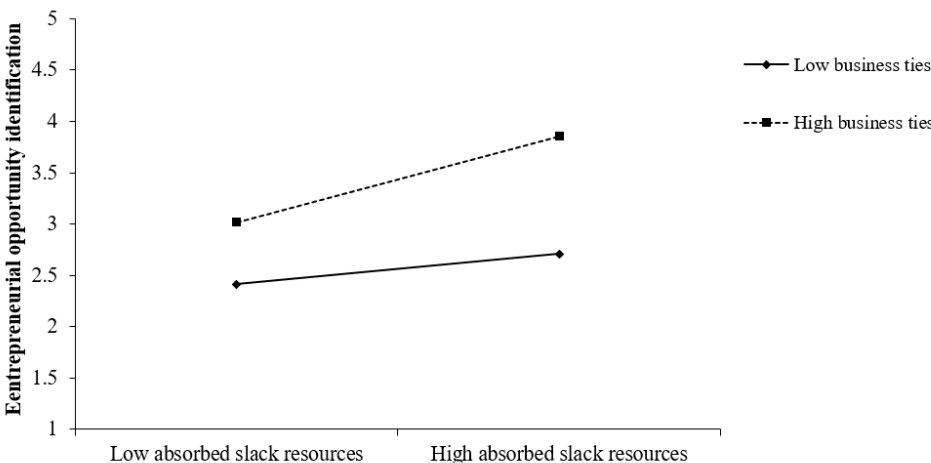

**Figure 2.** Moderating effect of business ties on absorbed slack resources and opportunity identification. Source: Excel drawing.

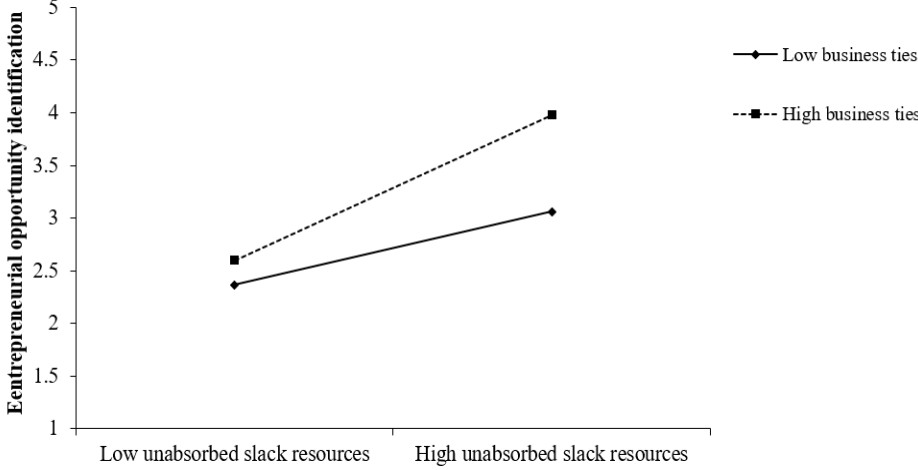

**Figure 3.** Moderating effect of business ties on unabsorbed slack resources and opportunity identification. Source: Excel drawing.

For the moderated mediation effect test of the main effect stage, this paper followed the test flow proposed by Edwards and Lambert (2007), and used Mplus 7.0 to test the moderated mediation effect of the main effect. The specific test procedure was establishing a regression equation for absorbed slack resources to resources bricolage and regression equations for slack resources, resource bricolage, business ties, and resource bricolage*business ties for entrepreneurial opportunity identification, specifically:

$$M = a_{0A} + a_{xA} \tag{1}$$

$$Y = b_{0A} + b_{xA}X + b_{mA}M + b_{zA}Z + b_{mzA}MZ \tag{2}$$

$$Y = [b_{0A} + b_{zA}Z + a_{0A}(b_{mA} + b_{nzA}Z)] + [b_{xA} + a_{xA}(b_{mA} + b_{mzA}Z)]X \tag{3}$$

$$M = a_{0U} + a_{xU} \tag{4}$$

$$Y = b_{0U} + b_{xU}X + b_{mU}M + b_{zU}Z + b_{mzU}MZ \tag{5}$$

$$Y = [b_{0U} + b_{zU}Z + a_{0U}(b_{mU} + b_{nzU}Z)] + [b_{xU} + a_{xU}(b_{mU} + b_{mzU}Z)]X \tag{6}$$

M stands for resources bricolage; $X_A$ stands for absorbed slack resources; $X_U$ stands for unabsorbed slack resources; Y stands for entrepreneurial opportunity identification; Z stands for business ties; $a_{0A}$, $a_{0U}$, $b_{0A}$, and $b_{0U}$ stand for constant coefficients in the corresponding regression equation, respectively; $a_{xA}$ and $_{xU}$ stand for regression coefficient of absorbed slack resources and unabsorbed slack resources on bricolage; $b_{XA} \backslash b_{xU}$, $b_m$, $b_z$, and $b_{mz}$ respectively stand for absorbed slack resources and unabsorbed slack resources, bricolage, business ties, and bricolage*business on entrepreneurial opportunity identification. First, substituting (1) into (2) gives (3), and substituting (4) into (5) gives (6). Second, using the Bootstrap method and taking 209 valid samples as the female parent, we randomly extracted 2000 new samples with a sample size of 209, and used Mplus7.0 to calculate the effect coefficient and its confidence interval; finally determining the significance of each effect and difference based on confidence intervals. The results of the Bootstrap test are shown in Tables 7 and 8.

**Table 7.** Moderated mediation effect (independent variable: Absorbed slack resources).

| Independent Variable | Moderator | Indirect Effect | Standard Error | 95% Confidence Interval |
|---|---|---|---|---|
| Absorbed slack resources | High business ties | 0.165 *** | 0.046 | [0.026, 0.179] |
| | Low business ties | 0.039 | 0.045 | [−0.001, 0.064] |
| | Difference | 0.126 * | 0.058 | [0.021, 0.193] |

Notes. *** $p < 0.001$, * $p < 0.05$.

**Table 8.** Moderated mediation effect (independent variable: Unabsorbed slack resources).

| Independent Variable | Moderator | Indirect Effect | Standard Error | 95% Confidence Interval |
|---|---|---|---|---|
| Unabsorbed slack resources | High business ties | 0.126 ** | 0.043 | [0.038, 0.203] |
| | Low business ties | 0.021 | 0.037 | [−0.005, 0.077] |
| | Difference | 0.105 * | 0.048 | [0.014, 0.189] |

Notes. ** $p < 0.01$, * $p < 0.05$.

As was shown in Table 7, business ties significantly moderated the mediating role of resources bricolage between absorbed slack resources and entrepreneurial opportunity identification. Specifically, under high business ties, the 95% confidence interval for indirect effects is [0.026, 0.197], excluding 0, r = 0.165, $p < 0.001$, indicating the impact of absorbed slack resources on entrepreneurial opportunity identification through resource bricolage was significant. Under low business ties, the 95% confidence interval for indirect effects is [−0.001, 0.064], including 0, r = 0.039, $p > 0.050$, indicating the impact of absorbed slack resources on entrepreneurial opportunity identification through resource bricolage was not significant. There was a significant difference in indirect effects between the two cases. The 95%

confidence interval for this difference was [0.021, 0.193], excluding 0, r = 0.126, *p* < 0.050. Therefore, H5a was supported.

As was shown in Table 8, business ties significantly moderated the mediating role of resources bricolage between unabsorbed slack resources and entrepreneurial opportunity identification. Specifically, under high business ties, the 95% confidence interval for indirect effects is [0.038, 0.203], excluding 0, r = 0.126, *p* < 0.010, indicating the impact of unabsorbed slack resources on entrepreneurial opportunity identification through resource bricolage was significant; under low business ties, the 95% confidence interval for indirect effects is [−0.005,0.077], including 0, r = 0.021, *p* > 0.050, indicating that the impact of unabsorbed slack resources on entrepreneurial opportunity identification through resource bricolage was not significant. There was a significant difference in indirect effects between the two cases. The 95% confidence interval of [0.014, 0.189], excluding 0, r = 0.105, *p* < 0.050. Therefore, H5b was supported.

Since the moderating role of political ties was not significant, political ties did not have a moderated mediation effect between slack resources (absorbed slack redundant and unabsorbed slack resources), resource bricolage, and entrepreneurial opportunity identification. H5c and H5d were not supported.

## 5. Discussion

### 5.1. General Discussion

On the basis of quantitative research conducted above, we drew the following conclusions:

First, slack resources have a positive impact on opportunity identification. Specifically, absorbed slack resources have a positive impact on entrepreneurial opportunity identification, and unabsorbed slack resources have a positive impact on entrepreneurial opportunity identification. This conclusion is consistent with previous research [13], emphasizing the positive effect of slack resources on entrepreneurial activities. On one hand, absorbed slack resources maintain stability and reduce internal conflicts. Previous studies have shown that absorbed slack resources are associated with utilizing opportunities [33]. Therefore, developable slack resources are advantageous for companies, especially in meeting opportunities and improving performance to promote growth. On the other hand, unabsorbed slack resources are easier to redeploy elsewhere, and this flexibility allows for greater management discretion. A management discretion is valuable to the organization's opportunity identification process, especially in China. Weak financial market infrastructure makes it difficult for enterprises to obtain resources directly from the market, emphasizing the development and creation of internal capital accumulation to find profitable opportunities in the market.

Second, resource bricolage plays a fully mediating role between the absorbed slack resources and opportunity identification. Resource bricolage plays a partial mediating role between the unabsorbed slack resources and opportunity identification. Additionally, resource bricolage has a greater mediating effect between absorbed slack resources and opportunity identification. Absorbed slack resources are the fixed cost of an enterprise. These resources have a certain rigidity and need to be combined by bricolage to trigger a resource allocation mechanism to create new knowledge or services. Additionally, the full mediating role shows the important role of resource bricolage. It can develop the value of them, use the only fixed absorbed slack resources, and use assets at hand to reconfigure for achieving new purposes to maximize the value of absorbed slack resources. Unabsorbed slack resources are a liquid asset of an enterprise with great flexibility. In the process of identifying entrepreneurial opportunities, companies need to find new ways to enhance the value of unabsorbed slack resources. Therefore, through the bricolage, we can continuously tap the potential of resources and enhance innovative utilization and reconfigure to achieve new goals by coordinating the effectiveness of resources.

Third, network ties play a moderating role between slack resources and opportunity identification. Specifically, commercial ties play a moderating role between the two types of slack resources and opportunity identification. As a kind of enterprise capital, business ties provide enterprises with access to market information. Enterprises can timely observe market demand, adopt targeted resource

allocation forms, and carry out activities such as service innovation, new product manufacturing, and new model creation. Then, enterprises can better amplify the effects of slack resources and make rational use of them through partnerships with suppliers and customers, to reduce risk costs, and make it easier to obtain market opportunities.

However, the moderating role of political ties between the two types of resources and opportunity identification is not significant. There have been studies that argue that social capital established by political ties can bring regulatory resources to firms. The results of this study are inconsistent with previous studies [63]. There may be three reasons for this. First, corporate managers have more relationships with business partners to develop their position and voice in the market, to obtain the latest relevant policy information, such as industrial development planning. However, this leads to a partial lack of political information, and does not fully understand the change dividend brought about by the policy. The role of political ties in identifying opportunities for enterprises using slack resources is not significant. Second, because of less contact with government departments and regulatory agencies, enterprises enjoy insufficient political legitimacy and are lacking in government protection in some aspects. Enterprises will encounter contractual or property rights disputes in the process of using slack resources to identify new opportunities. The weak political ties effect enterprises to explore profitable opportunities. Third, the protection of political ties is prone to the dependence of enterprises on political ties. They compromise the local government's goal of stabilizing the market or pursuing short-term political achievements, and lose the business objectives of the enterprise, which leads to weakening the exploration of new markets and meeting the status quo. So the ability to discover and identify opportunities is weakened to some extent.

Fourth, the model is a moderated mediation model. Specifically, business ties regulate the mediation role of resource bricolage in the relationship between the two types of slack resources and opportunity identification. By providing market resources, business ties enable enterprises to use slack resources to combined resources to find exploratory opportunities. Customer contact helps identify emerging customer and market needs, develop new distribution channels, and create new markets [64]. Good relationships with suppliers contributed to acquire new knowledge and develop new products by providing a new method [65]. Additionally, close contact with competitors contributed to promoting knowledge exchange and sharing [66]. These business ties provide opportunities for companies to integrate and restructure existing resources. Moreover, business ties make companies sensitive to environmental change, enabling companies to form improvisational creativity in environmental change to develop a spirit of exploration and trial and error, thus identifying and capturing new opportunities.

However, the moderated mediation effect of political ties between slack resources and opportunity identification is not significant. The reason why political ties cannot moderate the mediating role of resource bricolage is due to two aspects. On the one hand, entrepreneurs want to use the ties with the government to seek political asylum and access to the huge productive resources held by the government to enjoy more financing convenience and taxation. However, political ties have strong regulatory and strict specification boundaries. To maintain this relationship, enterprises must pay a certain cost, which leads enterprises to extract a considerable part from slack resources. Maintaining contact with the government has lost a lot of existing resources. Identifying entrepreneurial opportunities need to have enough resources at hand, whereas maintaining government ties weakens this part of the resources, resulting in a decline in the ability of the company's resource bricolage and new opportunity identification. On the other hand, entrepreneurs have spent some slack resources in the process of establishing business ties, which makes enterprises face the dilemma of insufficient resources when establishing ties with the government. They cannot acquire new political resources, reduce the cost of equity capital of enterprises, and enjoy less of the benefits of corporate loans and tax incentives [67]. In particular, the establishment of government ties is conducive to enhancing the legitimacy of enterprises in society and obtaining funding from regional institutions. Therefore, enterprises lacking political ties have lost their financial resources, which has limited the scope of their ability to combine their resources, and their ability to identify new opportunities is insufficient.

*5.2. Theoretical Implications*

Our study makes contributions to the literature in several ways:

First, introducing resource orchestration theory, this paper reveals the influence mechanism of slack resources on entrepreneurial opportunity identification, which makes up for the lack of resource-based theory and resource dependency theory, and enriches the literature on entrepreneurial opportunity identification. Resource-based view mainly emphasizes that high-quality resources can bring competitiveness to enterprises. However, when some companies have some resources that are not of high quality, they can combine rare resources to build competitive advantages. Resource dependence theory believed that companies rely on external resources and have positive interactions with the external stakeholders [6], but it does not explain how companies balance the relationship of both dependence and independence. To make up for the lack of two theories, this paper introduces the resource orchestration theory, emphasizing on the structuring, bundling, and leverage of resource elements [7], instead of paying attention to the merits of the resource attribute. Moreover, resource "smart" allocation embodies three functions of resource arrangement. Structuralizing involves "smart" acquisition, accumulation, and utilization of slack/network resources to form resource combinations. Bundling refers to "artful" resources, integrating and forming resource capacity, or steadily improving or expanding existing capacity, or creating new capacity. Improvisation and combination of resource bricolage to realize a new purpose is a kind of bundling of resources. Leverage emphasizes the "smart" use of capabilities to leverage opportunities, including mobilization, coordination, and deployment. Enterprise ensures the coordinated operation of the resource system to form competitiveness through the allocation and integration of three resource elements. The "smart" matching exactly responds to Sirmon's [8] recognition that value creation and competitive advantage development need to go hand in hand. At the same time, this article uses resource bricolage as a strategy and method for managers to integrate and utilize resources, and reveals the process of integrating internal slack resources and external resources to identify entrepreneurial opportunities through resource bricolage.

Second, previous studies have focused on the application of resource bricolage in the face of resource constraints. This paper expands this research, and believes that enterprises can also make resource bricolage based on certain slack resources, which expands the research situation of resource bricolage and enrich resource bricolage theory. The existing research mainly explores the use of resources bricolage to identify opportunity and enhance the competitiveness of enterprises [5,27]. However, the research neglects the situation that an enterprise has certain resources it can bricolage to identify opportunity. Therefore, this paper starts from the situation that the enterprise has certain slack resources, and deeply analyzes the internal mechanism of the identification of entrepreneurial opportunities through the integration and utilization of slack resources, which is conducive to promoting scholars to resource bricolage in the context of enterprises having resource base, and expands the research situation of resource bricolage.

Third, this study considers that network ties are a particularly important external network resource variable, and the enterprise interacts with partners in the external network relationship in which it embeds. Network ties can provide enterprises with external high-quality resources that are compatible with internal resources. Enterprises embedded in network can improve the flexibility and initiative of resource bricolage. When managers lie in different types of network ties, they can acquire policy and business resources, and those resources can promote the bricolage capability of companies, which will affect the corporate opportunity identification. Therefore, by introducing the external variables of network ties, the paper studies the influence mechanism of resource bricolage on the entrepreneurial opportunity identification based on internal and external resources under the different contexts, which extends research boundaries for entrepreneurial opportunity identification.

*5.3. Practical Implications*

The research in this paper also has implications for practice. First, correct strategic decisions are a prerequisite for the company's sustainable development. Enterprises should improve resource

bricolage capabilities on the basis of attributes of resource. On the one hand, enterprises should construct a unique ability to resource allocation by understanding the variety of elements. On the other hand, enterprises should find new collocation models by exploring the resource bricolage and using resource allocation capabilities to enhance their core competitiveness. They can cope with escalating risks and uncertainties and use new resource capabilities to seek new opportunities or applications for the exploration of new resources.

Second, to identify sustainable entrepreneurial opportunities, managers need to make good use of internal and external resources. First, managers need to properly utilize slack resources. Second, managers need to establish external network relationships. Network ties can not only enable enterprises to obtain high-quality external resources, but also capture mainstream information and obtain advanced resource bricolage methods. Finally, managers should strengthen cooperation with business partners and government agencies. Especially in the Chinese context, strengthening the relationship with the government is a very important corporate strategy. Through interaction and cooperation with the government, enterprises can obtain regulatory resources and policy resources, and reduce operating costs and operational risks to improve their competitive advantage.

*5.4. Limitations and Future Research*

Second, this paper considers the dynamics of resource management and introduces resource orchestration theory. However, see Sirmon et al. [57] for model verification. In fact, enterprises face multiple resource situations in real-life scenarios, and may use a variety of alternate or parallel resource bricolage methods for resource recombination or improvisation. Multi-dimensional resource bricolage measurement methods can better understand the type selection of resource bricolage methods, and through comparison and research, obtain the advantages and disadvantages of the selection results. Future research can use multi-dimensional measurement methods.

Second, this paper considers the dynamics of resource management and introduces resource orchestration theory. However, Sirmon et al. [8] believe that the life cycle factor of enterprises needs to be considered in the process of resource arrangement. Different life cycle stages present different resource management tools and effects, and identify different business opportunities. Future research can explore the differences in resource orchestration at each stage of the life cycle. For example, given that the resources available to companies in the start-up phase may differ from the resources that companies need to acquire or utilize at the maturity stage, research is needed to understand how entrepreneurs coordinate resource combinations from one phase to another.

Third, due to our limited costs, we have limited access to the top manager of the company. The sample size is relatively small, but the entire study design follows strict requirements. Future research may increase the sample size.

Finally, this paper explores the causality of resources to opportunities. Resources and opportunities exist as two independent factors and affect each other, and they also affect the entrepreneurial process at the same time. The two are interrelated and complementary. The complex relationship between resources and opportunities can be further combed in the future. For example, existing resources can lead to opportunity identification and development. The process of opportunity identification can be used to identify and acquire resources. In the path of resource allocation, resource identification can also be implemented to promote opportunity identification. Resources and opportunities are intertwined and fed back in complex systems to achieve the collaborative development of entrepreneurial activities.

## 6. Conclusions

This paper considers that the decision to deal with risk is to seek risk, that is, to find new opportunities. It may not bring benefits in the short term. From a long-term and sustainable perspective, new entrepreneurial opportunities can bring value premiums. From the perspective of resources-opportunity, integrated existing resources promote company opportunity identification. Based on the theory of resource orchestration, this paper establishes a process model from resource

acquisition to resource integration to opportunity identification, in order to answer the implementation process and result value of corporate behavioral decisions. In our analyses, we used data from the companies of eastern China, and statistical hypotheses were validated through a structural equation model. Several conclusions have been drawn. First, absorbed slack resources and unabsorbed slack resources have a positive impact on entrepreneurial opportunity identification. Second, slack resources indirectly affect the opportunity identification through the mediating role of resource bricolage. Third, business ties positively moderate the relationship between two types of slack resources and entrepreneurial opportunity identification, and business ties moderate the mediation effect of resource bricolage. The results provide practical implications for the top managers to identify opportunity by creatively utilizing internal and external resources. Measurement scales, construct items, measurement model and factor loading are shown in Appendix A.

**Author Contributions:** Conceptualized the research design, Y.S.; methodology, S.D. and Y.D.; software, Y.D.; validation, S.D.; formal analysis, S.D. and Y.D.; resources, Y.S.; data curation, S.D. and Y.D.; writing—original draft preparation, S.D. and Y.D.; writing—review and editing, Y.S.; visualization, Y.S.; supervision, Y.S.; project administration, Y.S.; funding acquisition, Y.S. All authors have read and agreed to the published version of the manuscript.

**Funding:** This research was supported by the National Social Science Foundation of China [Grant number 18BGL083].

**Conflicts of Interest:** The authors declare no conflict of interest.

## Appendix A

**Table A1.** Measurement scales and construct items.

| Items |
| --- |
| Slack resources |
| (1) Absorbed slack resources<br>    1.Our company's production capacity is not fully utilized |
|     2.Our company's equipment has not reached the limit of use |
|     3.Our company's operating level is lower than the production capacity and there is still a lot of room for development. |
| (2) Unabsorbed slack resources |
|     1. Our company has enough cash surplus |
|     2. Our company has a discretionary pool of funds |
|     3. Our company can get the bank loan we needed |
| Bricolage |
|     1. When faced with new challenges, our company is confident to use existing resources to find viable solutions. |
|     2. Compared with other companies, our company can use existing resources to cope with more challenges. |
|     3. Our company can effectively use any existing resources to deal with new problems or new opportunities in entrepreneurship. |
|     4. Our company can respond to new challenges by integrating existing resources and new resources that are cheaply available. |
|     5. When faced with new problems or opportunities, we usually assume that we can find a viable solution and tack action. |
|     6. By integrating existing resources, our company can effectively respond to any new challenges |
|     7. When faced with new challenges, our company can leverage existing resources to achieve viable solutions |
|     8. Our company can effectively address new challenges in the entrepreneurial process by integrating existing resources that were originally planned for other purposes. |

**Table A2.** Measurement scales and construct items.

| Items |
|---|
| Network ties |
| (1) Business ties |
|    1. Our company has a certain degree of commercial ties with suppliers. |
|    2. Our company has a certain degree of commercial ties with customers or distributors. |
|    3. Our company has a certain degree of commercial ties with competitor. |
| (2) Political ties |
|    1. Our company has a certain degree of political ties with the governments at all levels. |
|    2. Our company have a degree of political ties with industry associations |
|    3. Our company has a degree of political ties with the taxation department, business administration department and other management machines. |
| Opportunity identification |
|    1. Our company can quickly grasp the information of various entrepreneurial opportunities |
|    2. Our company can quickly identify the changes that new information may bring. |
|    3. The solution proposed by our company can match the target market demand very well. |

**Table A3.** Measurement model and factor loading.

| Construct | Items | Factor Loading | Cronbach's a | CR | AVE |
|---|---|---|---|---|---|
| **Absorbed slack resources** | AS1 | 0.734 | 0.897 | 0.800 | 0.571 |
| | AS2 | 0.781 | | | |
| | AS3 | 0.751 | | | |
| **Unabsorbed slack resources** | US1 | 0.744 | 0.892 | 0.810 | 0.588 |
| | US2 | 0.776 | | | |
| | US3 | 0.779 | | | |
| **Bricolage** | RB1 | 0.799 | 0.927 | 0.929 | 0.621 |
| | RB2 | 0.765 | | | |
| | RB3 | 0.765 | | | |
| | RB4 | 0.801 | | | |
| | RB5 | 0.801 | | | |
| | RB6 | 0.767 | | | |
| | RB7 | 0.785 | | | |
| | RB8 | 0.813 | | | |
| **Business ties** | BT1 | 0.714 | 0.923 | 0.825 | 0.612 |
| | BT2 | 0.796 | | | |
| | BT3 | 0.832 | | | |
| **Political ties** | PT1 | 0.756 | 0.891 | 0.815 | 0.595 |
| | PT2 | 0.758 | | | |
| | PT3 | 0.801 | | | |
| **Opportunity identification** | OI1 | 0.757 | 0.888 | 0.816 | 0.597 |
| | OI2 | 0.819 | | | |
| | OI3 | 0.740 | | | |

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
