# Peer review of "The Relationship between Slack Resources, Resource Bricolage, and Entrepreneurial Opportunity Identification—Based on Resource Opportunity Perspective"

_sustainability, doi:10.3390/su12031199_

Round 1
Reviewer 1 Report
Firstly, thank you very much for allowing me reading your work, and it is highly appreciated that you trust a reviewer’s credibility on the topic. I found your paper is interesting and indeed, it has a potential impact on the current debate on slack resources, resource bricolage and entrepreneurial opportunity identification.
Abstract section.
I would recommend to expend the methodology paragraph, and provide more insights on the used methods in the analysis.
For the theory and hypotheses section.
Although entrepreneurial opportunity identification is part of the article title, it’s treated rather expeditiously. There is a lot of emerging works on this topic but you can refer to the following work; Davidsson (2015), Vogel (2017), Foss and Klein (2018).
Material and methods
It would be interesting for the reader to know, what were the companies’ selection criteria? I understand that they were issued through research companies and projects but they are representative for their field of activity?. Second, no information is provided on where the study was applied. It was a survey at national/regional level, etc.
Third, the number of the respondents it’s quite low (209). More justification on this issue is needed.
Is it possible to adjust Table 3. Descriptive statistics of the variables and Table 6. The results of moderating effect. I can see inconsistency format here.
Overall: This is a good paper and has a potential impact. Please work on with the issues and concern raised, and if you have time, to consider the recommendations, I pointed out. With that, I wish you all the best and keep up good work!
Reviewer 2 Report
It was with great interest and satisfactions that I reviewed the paper. Well done.
The topic of the paper is current and relevant in the companies context.
The paper is well structured. The authors did an excellent literature review on the theories and keywords that guide the research, as well as the hypotheses are well formulated. The theoretical model accurately materializes the research guideline.
The methodology and statistical methods used are adequate to obtain the research objectives. The analysis of the results is detailed and correct. The authors could have been more ambitious in confronting their results with the results of other authors.
In may opinion, the text on lines 757, 758, 759, 762, 763, 764, 802, 803, 912, 913 should be revised.
Reviewer 3 Report
Please see attached my review.
